# Non-autoregressive Streaming Transformer for Simultaneous Translation

**Zhengrui Ma**[1,2], **Shaolei Zhang**[1,2], **Shoutao Guo**[1,2], **Chenze Shao**[1,2]
**Min Zhang**[3], **Yang Feng**[1,2*]

[1]Key Laboratory of Intelligent Information Processing
Institute of Computing Technology, Chinese Academy of Sciences
[2]University of Chinese Academy of Sciences
[3]School of Future Science and Engineering, Soochow University
{mazhengrui21b,fengyang}@ict.ac.cn  zhangminmt@hotmail.com

## Abstract

Simultaneous machine translation (SiMT) models are trained to strike a balance between latency and translation quality. However, training these models to achieve high quality while maintaining low latency often leads to a tendency for aggressive anticipation. We argue that such issue stems from the autoregressive architecture upon which most existing SiMT models are built. To address those issues, we propose non-autoregressive streaming Transformer (NAST) which comprises a unidirectional encoder and a non-autoregressive decoder with intra-chunk parallelism. We enable NAST to generate the blank token or repetitive tokens to adjust its READ/WRITE strategy flexibly, and train it to maximize the non-monotonic latent alignment with an alignment-based latency loss. Experiments on various SiMT benchmarks demonstrate that NAST outperforms previous strong autoregressive SiMT baselines. Source code is publicly available at https://github.com/ictnlp/NAST.

## 1 Introduction

Simultaneous machine translation (SiMT; Cho and Esipova, 2016; Gu et al., 2017; Ma et al., 2019; Arivazhagan et al., 2019; Zhang and Feng, 2023), also known as real-time machine translation, is commonly used in various practical scenarios such as live broadcasting, video subtitles and international conferences. SiMT models are required to start translation when the source sentence is incomplete, ensuring that listeners stay synchronized with the speaker. Nevertheless, translating partial source content poses significant challenges and increases the risk of translation errors. To this end, SiMT models are trained to strike a balance between latency and translation quality by dynamically determining when to generate tokens (i.e., WRITE action) and when to wait for additional source information (i.e., READ action).

However, achieving the balance between latency and translation quality is non-trivial for SiMT models. Training these models to produce high-quality translations while maintaining low latency often leads to a tendency for aggressive anticipation (Ma et al., 2019), as the models are compelled to output target tokens even before the corresponding source tokens have been observed during the training stage (Zheng et al., 2020). We argue that such an issue of anticipation stems from the autoregressive (AR) model architecture upon which most existing SiMT models are built. Regardless of the specific READ/WRITE strategy utilized, AR SiMT models are typically trained using maximum likelihood estimation (MLE) via teacher forcing. As depicted in Figure 1, their training procedure can have adverse effects on AR SiMT models in two aspects: 1) *non-monotonicity problem*: The reference used in training might be non-monotonically aligned with the source. However, in real-time scenarios, SiMT models are expected to generate translations that align monotonically with the source to reduce latency (He et al., 2015; Chen et al., 2021). The inherent verbatim alignment assumption during the MLE training of AR SiMT models restricts their performance; 2) *source-info leakage bias*: Following the practice in full-sentence translation systems, AR SiMT models deploy the teacher forcing strategy during training. However, it may inadvertently result in the leakage of source information. As illustrated in Figure 1, even if the available source content does not contain the word "举行 (hold)", the AR decoder is still fed with the corresponding translation word "held" as the ground truth context in training. This discrepancy between training and inference encourages the AR SiMT model to make excessively optimistic predictions during the real-time inference, leading to the issue of hallucination (Chen et al., 2021).

To address the aforementioned problems in autoregressive SiMT models, we focus on developing

---

*Corresponding author: Yang Feng

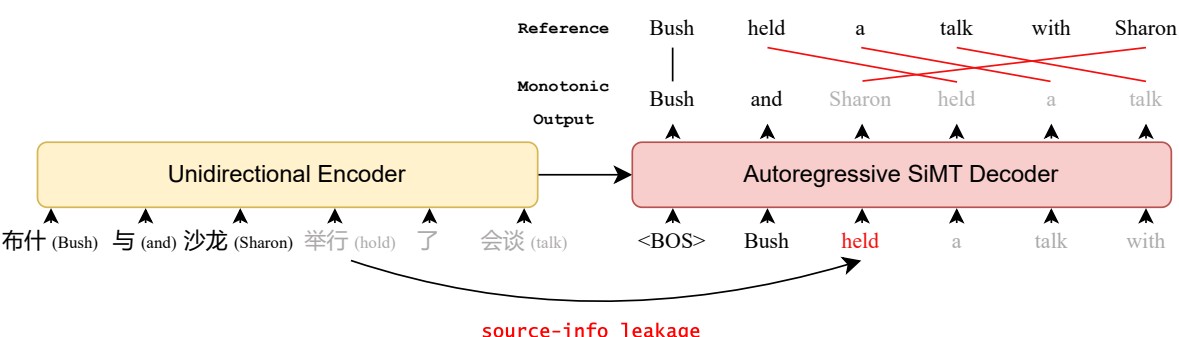

Figure 1: Illustration of the non-monotonicity problem and the source-info leakage bias in the training of autoregressive SiMT models. In this case, the AR SiMT model learns to predict at the third time step based on the source contexts "布什 (Bush)", "与 (and)", "沙龙 (Sharon)", and the ground truth contexts "Bush", "held". Although the source token "举行 (hold)" has not been read yet, it is exposed to the AR SiMT model through its corresponding token "held" in the ground truth context.

SiMT models that generate target tokens in a non-autoregressive (NAR) manner (Gu et al., 2018) by removing the target-side token dependency. We argue that an NAR decoder is better suited for streaming translation tasks. Firstly, the target tokens are modeled independently in NAR models, which facilitates the development of a non-monotonic alignment algorithm between generation and reference, alleviating the non-monotonicity problem. Additionally, the conditional independence assumption of the NAR structure liberates the model from the need for teacher forcing in training, thereby eliminating the risk of source-side information leakage. These advantageous properties of the NAR structure enable SiMT models to avoid aggressive anticipation and encourage the generation of monotonic translations with fewer reorderings that align with the output of professional human interpreters.

In this work, we propose non-autoregressive streaming Transformer (NAST). NAST processes streaming input and performs unidirectional encoding. Translations are generated in a chunk-by-chunk manner, with tokens within each chunk being generated in parallel. We enable NAST to generate blank token $\epsilon$ or repetitive tokens to build READ/WRITE paths adaptively, and train it to maximize non-monotonic latent alignment (Graves et al., 2006; Shao and Feng, 2022) with a further developed alignment-based latency loss. In this way, NAST effectively learns to generate translations that are properly aligned with the source in a monotonic manner, achieving high-quality translation while maintaining low latency.

Extensive experiments on WMT15 German → English and WMT16 English → Romanian bench-

marks demonstrate that NAST outperforms previous strong autoregressive SiMT baselines.

## 2 Preliminaries

### 2.1 Simultaneous Translation

Simultaneous machine translation models often adopt a prefix-to-prefix framework to start generating translation conditioned on partial source input. Given a source sentence $\boldsymbol{x} = \{x_1, ..., x_m\}$, previous autoregressive SiMT models factorize the probability of target sentence $\boldsymbol{y} = \{y_1, ..., y_n\}$ as:

$$p_g(\boldsymbol{y}|\boldsymbol{x}) = \prod_{t=1}^{|\boldsymbol{y}|} p(y_t|x_{\leq g(t)}, y_{<t}), \quad (1)$$

where $g(t)$ is a monotonic non-decreasing function of $t$, denoting the number of observed source tokens when generating $y_t$. A function $g(t)$ represents a specific READ/WRITE policy of SiMT models.

In addition to translation quality, latency is a crucial factor in the assessment of SiMT models. The latency of a policy $g(t)$ is commonly measured using Average Lagging (AL; Ma et al., 2019), which counts the number of tokens that the output lags behind the input:

$$AL(g; \boldsymbol{x}) = \frac{1}{\tau_g(|\boldsymbol{x}|)} \sum_{t=1}^{\tau_g(|\boldsymbol{x}|)} (g(t) - \frac{t-1}{r}), \quad (2)$$

where $\tau_g(|\boldsymbol{x}|)$ is the cut-off function to exclude the counting of problematic tokens at the end:

$$\tau_g(|\boldsymbol{x}|) = \min\{t|g(t) = |\boldsymbol{x}|\}, \quad (3)$$

and $r = \frac{|\boldsymbol{y}|}{|\boldsymbol{x}|}$ represents the length ratio between the target and source sequences.

### 2.2 Non-autoregressive Generation

#### 2.2.1 Parallel Decoding

Non-autoregressive generation (Gu et al., 2018) is originally introduced to reduce decoding latency[1]. It removes the autoregressive dependency and generates target tokens in a parallel way. Given a source sentence $\boldsymbol{x} = \{x_1, ..., x_m\}$, NAR models factorize the probability of target sentence $\boldsymbol{y} = \{y_1, ..., y_n\}$ as:

$$p(\boldsymbol{y}|\boldsymbol{x}) = \prod_{t=1}^{|\boldsymbol{y}|} p(y_t|\boldsymbol{x}). \qquad (4)$$

#### 2.2.2 Connectionist Temporal Classification

Unlike autoregressive models that dynamically control the length by generating the <eos> token, NAR models often utilize a length predictor to pre-determine the length of the output sequence before generation. The predicted length may be imprecise and lacks adaptability for adjustment. Connectionist Temporal Classification (CTC; Graves et al., 2006) addresses this limitation by extending the output space $\mathcal{Y}$ with a blank token $\epsilon$. The generation $\boldsymbol{a} \in \mathcal{Y}^*$ is referred to as the alignment. CTC defines a mapping function $\beta(\boldsymbol{y}; T)$ that returns a set of all possible alignments of $\boldsymbol{y}$ of length $T$ and a collapsing function $\beta^{-1}(\boldsymbol{a})$ that first collapses all consecutive repeated tokens in $\boldsymbol{a}$ and then removes all blanks to obtain the target. During training, CTC marginalizes out all alignments:

$$p(\boldsymbol{y}|\boldsymbol{x}) = \sum_{\boldsymbol{a} \in \beta(\boldsymbol{y};T)} p(\boldsymbol{a}|\boldsymbol{x}), \qquad (5)$$

where $T$ is a pre-determined length and the alignment is modeled in a non-autoregressive way:

$$p(\boldsymbol{a}|\boldsymbol{x}) = \prod_{t=1}^{T} p(a_t|\boldsymbol{x}). \qquad (6)$$

## 3 Approach

We provide a detailed introduction to the non-autoregressive streaming Transformer (NAST) in this section.

### 3.1 Architecture Overview

NAST consists of a unidirectional encoder (Arivazhagan et al., 2019; Ma et al., 2019; Miao et al.,

---

[1]Note that the concept of *latency* differs between NAR generation and SiMT. It refers to the delay in generating all target tokens once all source tokens are observed in the first case and to the level of synchronization between target-side generation and source-side observation in the latter case.

---

2021) and a non-autoregressive decoder with intra-chunk parallelism. The model architecture is depicted in Figure 2. When a source token $\boldsymbol{x}_i$ is read in, NAST passes it to the unidirectional encoder, allowing it to attend to the previous source contexts through causal encoder self-attention:

$$\text{SelfAttn}(\boldsymbol{x}_i, \boldsymbol{x}_{\leq i}). \qquad (7)$$

Concurrently, NAST upsamples $\boldsymbol{x}_i$ $\lambda$ times and feeds them to construct the decoder hidden states as a chunk. Within the chunk, NAST handles $\lambda$ states in a fully parallel manner. To further clarify, we introduce $\boldsymbol{h}$ to represent the sequence of decoder states. Thus, the $j$-th hidden state in the $i$-th chunk can be denoted as $\boldsymbol{h}_{(i-1)\lambda+j}$, subject to $1 \leq i \leq |\boldsymbol{x}|$ and $1 \leq j \leq \lambda$. Those states can attend to information from all currently observed source contexts through cross-attention:

$$\text{CrossAttn}(\boldsymbol{h}_{(i-1)\lambda+j}, \boldsymbol{x}_{\leq i}), \qquad (8)$$

and to information from all constructed decoder states through self-attention:

$$\text{SelfAttn}(\boldsymbol{h}_{(i-1)\lambda+j}, \boldsymbol{h}_{\leq i\lambda}). \qquad (9)$$

Following CTC (Graves et al., 2006), we extend the vocabulary to allow NAST generating the blank token $\epsilon$ or repeated tokens from decoder states to model an implicit READ action. We refer to the outputs from a states chunk $\boldsymbol{h}_{(i-1)\lambda+1:i\lambda}$ as partial alignments $\boldsymbol{a}_{(i-1)\lambda+1:i\lambda}$, where NAST generates them in a non-autoregressive way:

$$p(\boldsymbol{a}_{(i-1)\lambda+1:i\lambda}|\boldsymbol{h}_{(i-1)\lambda+1:i\lambda})$$
$$= \prod_{j=1}^{\lambda} p(\boldsymbol{a}_{(i-1)\lambda+j}|\boldsymbol{h}_{(i-1)\lambda+j}). \qquad (10)$$

To obtain the translation stream, we first apply the collapsing function $\beta^{-1}$ to deal with the partial alignments generated from the $i$-th chunk:

$$\boldsymbol{y}^{\text{chunk}_i} = \beta^{-1}(\boldsymbol{a}_{(i-1)\lambda+1:i\lambda}). \qquad (11)$$

Then NAST concatenates the outputs from the current chunk to generated prefix $\boldsymbol{y}^{\text{pre}}$ according to the following rule:

$$\begin{cases} \boldsymbol{y}^{\text{pre}} = \boldsymbol{y}^{\text{pre}} \oplus \boldsymbol{y}^{\text{chunk}_i}_{2:}, \text{ if } \boldsymbol{y}^{\text{pre}}_{-1} = \boldsymbol{y}^{\text{chunk}_i}_1 \\ \boldsymbol{y}^{\text{pre}} = \boldsymbol{y}^{\text{pre}} \oplus \boldsymbol{y}^{\text{chunk}_i}, \text{ otherwise} \end{cases} \qquad (12)$$

where $\boldsymbol{y}^{\text{pre}}_{-1}$ denotes the last token in the generated prefix. Consequently, upon receiving a token in the input stream, NAST is capable to generate 0 to $\lambda$

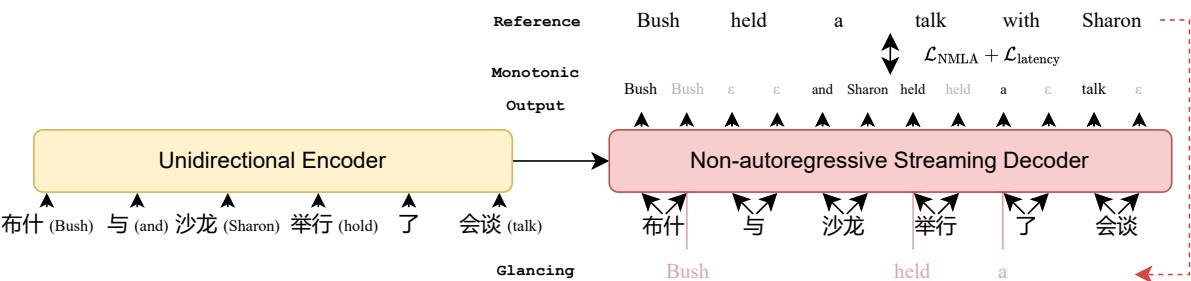

Figure 2: Overview of the proposed non-autoregressive streaming Transformer (NAST). Upon receiving a source token, NAST upsamples it $\lambda$ times and feeds them to the decoder as a chunk. NAST can generate blank token $\epsilon$ or repetitive tokens (both highlighted in gray) to find reasonable READ/WRITE paths adaptively. We train NAST using the non-monotonic latent alignment loss (Shao and Feng, 2022) with the alignment-based latency loss to achieve translation of high quality while maintaining low latency.

tokens at a time, endowing it with the ability to adjust its READ/WRITE strategy flexibly. Formally, each full alignment $\boldsymbol{a} \in \beta(\boldsymbol{y}; \lambda|\boldsymbol{x}|)$ can be considered as a concatenation of all the partial alignments, and implies a specific READ/WRITE policy to generate the reference $\boldsymbol{y}$. Therefore, NAST jointly models the distribution of translation and READ/WRITE policy by marginalizing out latent alignments:

$$
\begin{aligned}
p(\boldsymbol{y}|\boldsymbol{x}) &= \sum_{\boldsymbol{a} \in \beta(\boldsymbol{y}; \lambda|\boldsymbol{x}|)} p(\boldsymbol{a}|\boldsymbol{x}) \\
&= \sum_{\boldsymbol{a} \in \beta(\boldsymbol{y}; \lambda|\boldsymbol{x}|)} \prod_{\substack{1 \leq i \leq |\boldsymbol{x}| \\ 1 \leq j \leq \lambda}} p(\boldsymbol{a}_{(i-1)\lambda+j}|\boldsymbol{x}_{\leq i}).
\end{aligned} \tag{13}
$$

### 3.2 Latency Control

While NAST exhibits the ability to adaptively determine an appropriate READ/WRITE policy, we want to impose some specific requirements on the trade-off between latency and translation quality. To accomplish this, we introduce an alignment-based latency loss and a chunk wait-$k$ strategy to effectively control the latency of NAST.

#### 3.2.1 Alignment-based Latency Loss

Considering NAST models the distribution of READ/WRITE policy by capturing the distribution of latent alignments, it is desirable to measure the averaged latency of all latent alignments and further regularize it. Specifically, we are interested in the expected Average Lagging (AL; Ma et al., 2019) of NAST:

$$
AL(\theta; \boldsymbol{x}) = \mathbb{E}_{\boldsymbol{a} \sim p_\theta(\boldsymbol{a}|\boldsymbol{x})}[AL(g^{\boldsymbol{a}}; \boldsymbol{x})], \tag{14}
$$

where $g^{\boldsymbol{a}}$ is the policy induced from alignment $\boldsymbol{a}$. Due to the exponentially large alignment space, it

is infeasible to enumerate all possible $g^{\boldsymbol{a}}$ to obtain $AL(\theta; \boldsymbol{x})$. This limitation motivates us to delve deeper into $AL(\theta; \boldsymbol{x})$ and devise an efficient estimation algorithm.

To simplify the estimation process of $AL(\theta; \boldsymbol{x})$ while still excluding the lag counting of problematic words generated after all source read in, we deploy a new cut-off function that disregards tokens generated after all source observed, i.e., tokens from the last chunk:

$$
\tau_{g^{\boldsymbol{a}}}(|x|) = \min\{t|g^{\boldsymbol{a}}(t) = |x|\} - 1. \tag{15}
$$

Then we introduce a moment function $m(i)$ to denote the number of observed source tokens when generating the $i$-th position in the alignment. Given the fixed upsampling strategy of NAST, it is clear that:

$$
m((i-1)\lambda+j) = i, \ 1 \leq j \leq \lambda. \tag{16}
$$

We further define an indicator function $\mathbb{1}(\boldsymbol{a}_i)$ to denote whether the $i$-th position in the alignment is reserved after collapsed by $\beta^{-1}$. With its help, it is convenient to express the lagging of alignment $\boldsymbol{a}$:

$$
\begin{aligned}
&AL(g^{\boldsymbol{a}}; \boldsymbol{x}) \\
&= \frac{1}{\tau_{g^{\boldsymbol{a}}}(|\boldsymbol{x}|)} \left( \sum_{t=1}^{\tau_{g^{\boldsymbol{a}}}(|\boldsymbol{x}|)} g(t) - \sum_{t=1}^{\tau_{g^{\boldsymbol{a}}}(|\boldsymbol{x}|)} \frac{t-1}{r} \right) \\
&= \frac{1}{\tau_{g^{\boldsymbol{a}}}(|\boldsymbol{x}|)} \left( \sum_{i=1}^{(|\boldsymbol{x}|-1)\lambda} m(i)\mathbb{1}(\boldsymbol{a}_i) - \frac{\tau_{g^{\boldsymbol{a}}}(|\boldsymbol{x}|)(\tau_{g^{\boldsymbol{a}}}(|\boldsymbol{x}|)-1)}{2r} \right) \\
&\approx \frac{1}{\tau_{g^{\boldsymbol{a}}}(|\boldsymbol{x}|)} \left( \sum_{i=1}^{(|\boldsymbol{x}|-1)\lambda} m(i)\mathbb{1}(\boldsymbol{a}_i) - \frac{|\boldsymbol{x}|(\tau_{g^{\boldsymbol{a}}}(|\boldsymbol{x}|)-1)}{2} \right).
\end{aligned} \tag{17}
$$

Equation 17 inspires us to estimate the expected average lagging $AL(\theta; \boldsymbol{x})$ by separately calculating the expected values of the numerator and denomi-

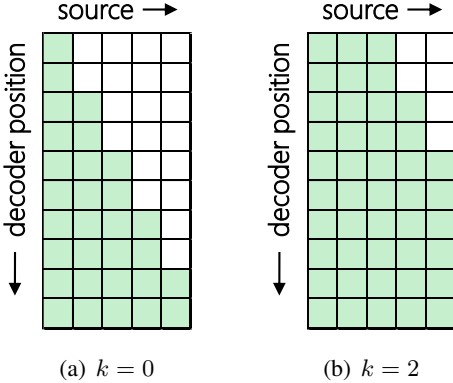

(a) $k = 0$       (b) $k = 2$

Figure 3: Illustration of cross-attention with different chunk wait-$k$ strategies.

nator:

$$AL(\theta; \boldsymbol{x})$$
$$\approx \frac{\mathbb{E}_{\boldsymbol{a}}[\sum_{i=1}^{(|\boldsymbol{x}|-1)\lambda} m(i)\mathbb{1}(\boldsymbol{a}_i)] - \frac{|\boldsymbol{x}|}{2}(\mathbb{E}_{\boldsymbol{a}}[\tau_{g^{\boldsymbol{a}}}(|\boldsymbol{x}|)] - 1)}{\mathbb{E}_{\boldsymbol{a}}[\tau_{g^{\boldsymbol{a}}}(|\boldsymbol{x}|)]}.$$
(18)

It relieves us from the intractable task of enumerating $g^{\boldsymbol{a}}$. Instead, we only need to handle two expectation terms: $\mathbb{E}_{\boldsymbol{a}}[\sum_{i=1}^{(|\boldsymbol{x}|-1)\lambda} m(i)\mathbb{1}(\boldsymbol{a}_i)]$ and $\mathbb{E}_{\boldsymbol{a}}[\tau_{g^{\boldsymbol{a}}}(|\boldsymbol{x}|)]$, which can be resolved efficiently:[2]

$$\begin{cases} \mathbb{E}_{\boldsymbol{a}}[\tau_{g^{\boldsymbol{a}}}(|\boldsymbol{x}|)] = \sum_{i=1}^{(|\boldsymbol{x}|-1)\lambda} p(\mathbb{1}(\boldsymbol{a}_i)) \\ \mathbb{E}_{\boldsymbol{a}}[\sum_{i=1}^{(|\boldsymbol{x}|-1)\lambda} m(i)\mathbb{1}(\boldsymbol{a}_i)] = \sum_{i=1}^{(|\boldsymbol{x}|-1)\lambda} m(i)p(\mathbb{1}(\boldsymbol{a}_i)) \end{cases}$$
(19)

where $p(\mathbb{1}(\boldsymbol{a}_i))$ represents the probability that the $i$-th token in the alignment is reserved after collapsing and can be calculated simply as:

$$p(\mathbb{1}(\boldsymbol{a}_i)) = 1 - p(\boldsymbol{a}_i = \epsilon) - \sum_{v \in \mathcal{Y}/\epsilon} p(\boldsymbol{a}_i = v)p(\boldsymbol{a}_{i-1} = v).$$
(20)

With the assistance of the aforementioned derivation, it is efficient to estimate the expected average lagging of NAST. By applying it along with a tunable minimum lagging threshold $l_{\min}$, we can train NAST to meet specific requirements of low latency:

$$\mathcal{L}_{\text{latency}} = \max(AL(\theta; \boldsymbol{x}), l_{\min}).$$
(21)

### 3.2.2 Chunk Wait-$k$ Strategy

In addition to the desiring property of shorter lagging, there may be practical scenarios where we aim to mitigate the risk of erroneous translations by increasing the latency. To this end, we propose a chunk wait-$k$ strategy for NAST to satisfy the requirements of better translation quality.

---

[2]We leave the detailed derivation of Equation 19 in Appendix A.

NAST is allowed to wait for additional $k$ source tokens before initializing the generation of the first chunk. The first chunk is fed to the decoder at the moment the $(k + 1)$-th source token is read in. Subsequently, NAST feeds each following chunk as each new source token is received. The partial alignment generated from each chunk is consistently lagged by $k$ tokens compared with the corresponding source token until the source sentence is complete.

Formally, the moment function for the chunk wait-$k$ strategy can be formulated as:

$$m((i-1)\lambda + j) = \min\{i + k, |\boldsymbol{x}|\}, \ 1 \le j \le \lambda.$$
(22)

As depicted in Figure 3, decoder states can further access information from additional $k$ observed source tokens through cross-attention:

$$\text{CrossAttn}(\boldsymbol{h}_{(i-1)\lambda+j}, \boldsymbol{x}_{\le \min\{i+k,|\boldsymbol{x}|\}}),$$
(23)

which leads NAST to prioritize better translation quality at the expense of longer delay.

### 3.3 Non-monotonic Latent Alignments

While CTC loss (Graves et al., 2006) provides the convenience of directly applying the maximum likelihood estimation to train NAST, i.e., $\mathcal{L} = -\log p(\boldsymbol{y}|\boldsymbol{x})$, it only considers the monotonic mapping from target positions to alignment positions. However, non-monotonic alignments are crucial in simultaneous translation. SiMT models are expected to generate translations that are monotonically aligned with the source sentence to achieve low latency. Unfortunately, in the training corpus, source and reference pairs are often non-monotonically aligned due to differences in grammar structures between languages (e.g., SVO vs SOV). Neglecting the non-monotonic mapping during training compels the model to predict tokens for which the corresponding source has not been read, resulting in over-anticipation. To address these issues, we apply the bigram-based non-monotonic latent alignment loss (Shao and Feng, 2022) to train our NAST, which maximizes the F1 score of expected bigram matching between target and alignments:

$$\mathcal{L}_{\text{NMLA}}(\theta) = -\frac{2 \cdot \sum_{g \in G_2} \min\{C_g(y), C_g(\theta)\}}{\sum_{g \in G_2}(C_g(y) + C_g(\theta))},$$
(24)

where $C_g(y)$ denotes the occurrence count of bigram $g = (g_1, g_2)$ in the target, $C_g(\theta)$ represents the expected count of $g$ for NAST, and $G_2$ denotes the set of all bigrams in $y$.

### 3.4 Glancing

Due to its inherent conditional independence structure, NAST may encounter challenges related to the multimodality problem[3] (Gu et al., 2018). To address this issue, we employ the glancing strategy (Qian et al., 2021) during training. This involves randomly replacing tokens in the decoder's input chunk with tokens from the most probable latent alignment. Formally, the glancing alignment is the one that maximizes the posterior probability:

$$a^* = \underset{a \in \beta(y;\lambda|x|)}{\arg\max}\ p(a|x). \tag{25}$$

Then we randomly sample some positions in the decoder input and replace tokens in the input sequence with tokens from the glancing alignment sequence at those positions in training.

### 3.5 Training Strategy

In order to better train the NAST model to adapt to simultaneous translation tasks with different latency requirements, we propose a two-stage training strategy. In the first stage, we train NAST using the CTC loss to obtain the reference monotonic-aligned translation with adaptive latency:

$$\mathcal{L}_{stage-1} = \mathcal{L}_{\text{CTC}} = -\log p(y|x). \tag{26}$$

In the second stage, we train NAST using the combination of the non-monotonic latent alignment loss and the alignment-based latency loss:

$$\mathcal{L}_{stage-2} = \mathcal{L}_{\text{NMLA}} + \mathcal{L}_{\text{latency}}. \tag{27}$$

This further enables NAST to generate translations that are aligned with the source in a monotonic manner, meeting specific latency requirements.

## 4 Experiments

### 4.1 Experimental Setup

**Datasets** We conduct experiments on the following benchmarks that are widely used in previous SiMT studies: WMT15[4] German → English (De→En, 4.5M pairs) and WMT16[5] English → Romanian (En→Ro, 0.6M pairs). For De→En, we use *newstest2013* as the validation set and *newstest2015* as the test set. For En→Ro, we use *newsdev-2016*

as the validation set and *newstest-2016* as the test set. For each dataset, we apply BPE (Sennrich et al., 2016) with 32k merge operations to learn a joint subword vocabulary shared across source and target languages.

**Implementation Details** We select a chunk upsample ratio of 3 ($\lambda = 3$) and adjust the chunk waiting parameter $k$ and the threshold $l_{\min}$ in alignment-based latency loss to achieve varying quality-latency trade-offs.[6] For the first stage of training, we set the dropout rate to 0.3, weight decay to 0.01, and apply label smoothing with a value of 0.01. We train NAST for 300k updates on De→En and 100k updates on En→Ro. A batch size of 64k tokens is utilized, and the learning rate warms up to $5 \cdot 10^{-4}$ within 10k steps. The glancing ratio linearly anneals from 0.5 to 0.3 within 200k steps on De→En and 100k steps on En→Ro. In the second stage, we apply the latency loss only if the chunk wait strategy is disabled ($k = 0$). The dropout rate is adjusted to 0.1 for De→En, while no label smoothing is applied to either task. We further train NAST for 10k updates on De→En and 6k updates on En→Ro. A batch size of 256k tokens is utilized to stabilize the gradients, and the learning rate warms up to $3 \cdot 10^{-4}$ within 500 steps. The glancing ratio is fixed at 0.3. During both training stages, all models are optimized using Adam (Kingma and Ba, 2014) with $\beta = (0.9, 0.98)$ and $\epsilon = 10^{-8}$. Following the practice in previous research on non-autoregressive generation, we employ sequence-level knowledge distillation (Kim and Rush, 2016) to reduce the target-side dependency in data.[7] We adopt Transformer-base (Vaswani et al., 2017) as the offline teacher model and train NAST on the distilled data.

**Baselines** We compare our system with the following strong autoregressive SiMT baselines:

**Offline AT** Transformer model (Vaswani et al., 2017), which initiates translation after reading all the source tokens. We utilize a unidirectional encoder and employ greedy search decoding for fair comparison.

---

[3] The *multimodality problem* arises when a source sentence has multiple possible translations, which a non-autoregressive system is unable to capture due to its inability to model the target dependency.

[4] https://www.statmt.org/wmt15/

[5] https://www.statmt.org/wmt16/

[6] Further details regarding the settings of $k$ and $l_{\min}$ can be found in Appendix B.

[7] Note that the purpose of offline knowledge distillation is to reduce the dependency between target-side tokens in the data, in order to facilitate the learning of non-autoregressive models. This is different from the goal of performing monotonic knowledge distillation in the field of SiMT, which aims to obtain monotonic aligned data.

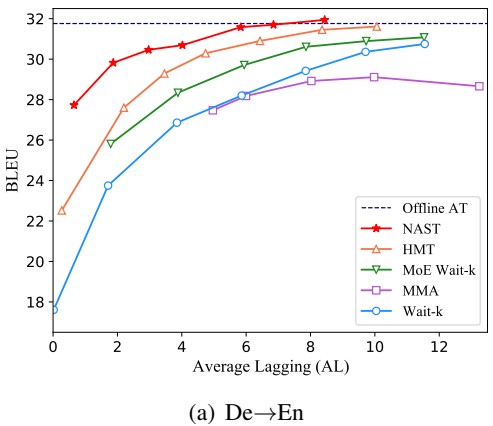

(a) De→En

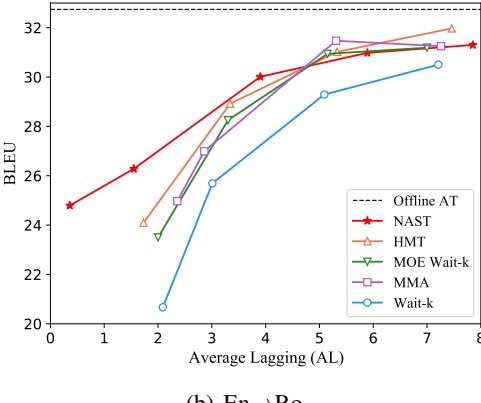

(b) En→Ro

Figure 4: Results of translation quality (BLEU) against latency (Average Lagging) on De→En and En→Ro.

**Wait-$k$** Wait-$k$ policy (Ma et al., 2019), which initially reads $k$ tokens and subsequently alternates between WRITE and READ actions.

**MoE Wait-$k$** Mixture-of-experts wait-$k$ policy (Zhang and Feng, 2021), which involves employing multiple experts to learn multiple wait-$k$ policies during training. MoE Wait-$k$ is the current SOTA fixed policy.

**MMA** Monotonic multi-head attention (MMA; Ma et al., 2020) employs a Bernoulli variable to predict the READ/WRITE action and is trained using monotonic attention (Raffel et al., 2017).

**HMT** Hidden Markov Transformer (HMT; Zhang and Feng, 2023), which treats the moments of starting translating as hidden events and considers the target sequence as the observed events. This approach organizes them as a hidden Markov model. HMT is the current SOTA adaptive policy.

**Metrics** To compare SiMT models, we evaluate the translation quality using BLEU score (Papineni et al., 2002) and measure the latency using Average Lagging (AL; Ma et al., 2019). Numerical results with more latency metrics can be found in Appendix B.

## 4.2 Main Results

We compare NAST with the existing AR SiMT methods in Figure 4. On De→En, NAST outperforms all AR SiMT models significantly across all latency settings, particularly in scenarios with very low latency. With the latency in the range of $[0, 1]$, where listeners are almost synchronized with the speaker, NAST achieves a translation quality of 27.73 BLEU, surpassing the current SOTA model HMT by nearly 6 BLEU points. Moreover, NAST

|  |  | $k$ | 0 | 3 | 5 | 7 |
|---|---|---|---|---|---|---|
| **NAST** | **BLEU** | | 30.69 | 31.58 | 31.70 | 31.94 |
| **w/o $\mathcal{L}_{\text{NMLA}}$** | | | 28.84 | 29.72 | 30.12 | 30.68 |
|  | $\Delta$ | | 1.85 | 1.86 | 1.58 | 1.26 |

Table 1: Results of BLEU scores on De→En test set with or without $\mathcal{L}_{\text{NMLA}}$ under different chunk wait-$k$ strategy. $\mathcal{L}_{\text{latency}}$ is not applied here.

|  |  | $k$ | 0 | 3 | 5 | 7 |
|---|---|---|---|---|---|---|
| **NAST** | **AL** | | 4.02 | 5.83 | 6.85 | 8.44 |
| **w/o $\mathcal{L}_{\text{NMLA}}$** | | | 3.42 | 5.11 | 6.56 | 8.20 |

Table 2: Results of Average Lagging on De→En test set with or without $\mathcal{L}_{\text{NMLA}}$ under different chunk wait-$k$ strategy. $\mathcal{L}_{\text{latency}}$ is not applied here.

demonstrates superior performance compared to the offline AT system even when the AL is as low as 6.85, showcasing its competitiveness in scenarios where higher translation quality is desired. On En→Ro, NAST also exhibits a substantial improvement under low latency conditions. On the other hand, NAST achieves comparable performance to other models on En→Ro when the latency requirement is not stringent.

## 5 Analysis

### 5.1 Importance of Non-monotonic Alignments

NAST is trained using a non-monotonic alignment loss, enabling it to generate source monotonic-aligned translations akin to human interpreters. This capability empowers NAST to achieve high-quality streaming translations while maintaining low latency. To validate the effectiveness of non-monotonic alignment, we conduct further experi-

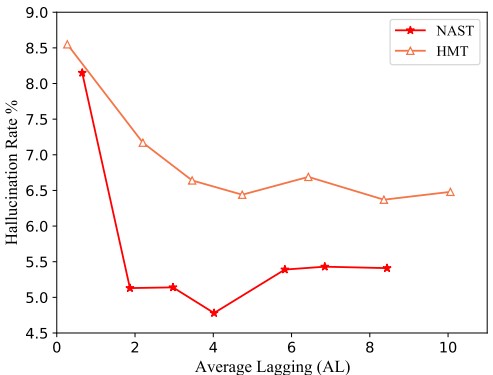

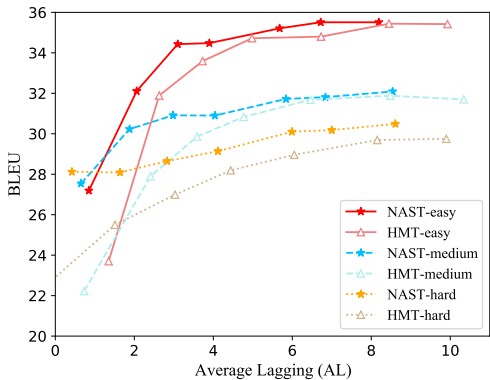

Figure 5: Results of hallucination rate against latency (Average Lagging) on De→En test set.

Figure 6: Performance on De→En test subsets categorized by difficulty.

ments by studying the performance of NAST without $\mathcal{L}_{\text{NMLA}}$. We compare the translation quality (BLEU) and latency (AL) of models employing different chunk wait-$k$ strategies. The results are reported in Table 1 and Table 2. Note that $\mathcal{L}_{\text{latency}}$ is not applied here for clear comparison.

We observe that incorporating $\mathcal{L}_{\text{NMLA}}$ significantly enhances translation quality by up to 1.86 BLEU, while maintaining nearly unchanged latency. We also notice that the improvement is particularly substantial when the latency is low, which aligns with our motivation. Under low latency conditions, SiMT models face more severe non-monotonicity problems. The ideal simultaneous generation requires more reordering of the reference to achieve source sentence monotonic alignment, which leads to greater improvements of applying non-monotonic alignment loss.

### 5.2 Analysis on Hallucination Rate

NAST mitigates the risk of source information leakage during training, thereby minimizing the occurrence of hallucination during inference. To demonstrate this, we compare the hallucination rate (Chen et al., 2021) of hypotheses generated by NAST with that of the current SOTA model, HMT (Zhang and Feng, 2023). A hallucination is defined as a generated token that can not be aligned to any source word. The results are plotted in Figure 5.

We note that the hallucination rates of both models decrease as the latency increases. However, NAST exhibits a significantly lower hallucination rate compared to HMT. We attribute this to the fact that NAST avoids the bias caused by source-info leakage and enables a more general generation-reference alignment, thus mitigating compelled predictions during training.

### 5.3 Performance across Difficulty Levels

To further illustrate NAST's effectiveness in handling non-monotonicity problem, we investigate its performance when confronted with samples of varying difficulty levels. It is intuitive to expect that samples with a higher number of cross alignments between the source and reference texts pose a greater challenge for real-time translation. Therefore, we evenly partition the De→En test set into subsets based on the number of crosses in the alignments, categorizing them as easy, medium, and hard, in accordance with the approach by Zhang and Feng (2021). We compare our NAST with previous HMT model, and the results are presented in Figure 6.

Despite the impressive performance of NAST, a closer examination of Figure 6 reveals that the superiority is particular on the challenging subset. Even when real-time requirements are relatively relaxed, the improvement in handling the hard subset remains noteworthy. We attribute this to the stringent demand imposed by the hard subset, requiring SiMT models to effectively manage word reorderings to handle the non-monotonicity. NAST benefits from non-monotonic alignment training and excels in addressing these challenges, thus enhancing its performance in handling those harder samples.

### 5.4 Concerns on Fluency

While the non-autoregressive nature endows NAST with the capability to tackle the non-monotonicity problem and source-info leakage bias, it also exposes NAST to the risk of potential fluency degradation due to the absence of target-side dependency. To have a better understanding of this problem, we

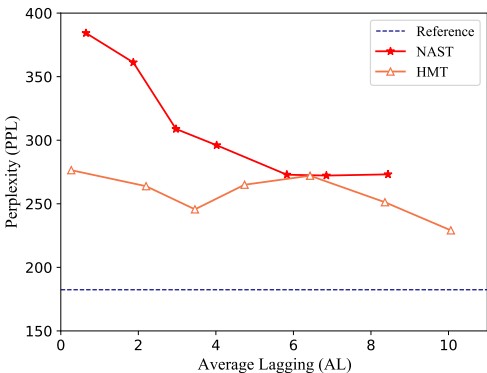

Figure 7: Results of fluency (Perplexity) against latency (Average Lagging) on De→En test set.

evaluate the fluency of the De→En test set output from NAST in comparison to previous HMT. Specifically, we employ the Perplexity value reported by an external pre-trained language model `transformer_lm.wmt19.en`[8] to measure the fluency of the generated texts. A lower Perplexity value implies more fluent translations. The results are presented in Figure 7.

Though NAST exhibits significantly improved translation quality, we find its non-autoregressive nature does impact fluency to some extent. However, we consider this trade-off acceptable. In practical scenarios like international conferences where SiMT models are employed, the language used by human speakers is often not perfectly fluent. In such contexts, the audience tends to prioritize the overall translation quality under low latency, rather than the fluency of generated sentences.

## 6  Related Work

**SiMT** Simultaneous machine translation requires a READ/WRITE policy to balance latency and translation quality, involving fixed and adaptive strategies. For the fixed policy, Ma et al. (2019) proposed wait-$k$, which first reads $k$ source tokens and then alternates between READ/WRITE action. Elbayad et al. (2020) introduced an efficient training method for the wait-$k$ policy, which randomly samples $k$ during training. Zhang and Feng (2021) proposed a mixture-of-experts wait-$k$ to learn a set of wait-$k$ policies through multiple experts. For the adaptive policy, Gu et al. (2017) trained an agent to decide READ/WRITE via reinforcement learning. Arivazhagan et al. (2019) introduced MILk, which incorporates a Bernoulli variable to indicate the

[8] https://github.com/facebookresearch/fairseq/tree/main/examples/language_model

READ/WRITE action. Ma et al. (2020) proposed MMA to implement MILk on Transformer. Liu et al. (2021) introduced CAAT, which leverages RNN-T and employs a blank token to signify the READ action. Miao et al. (2021) proposed GSiMT to generate the READ/WRITE actions. Chang et al. (2022) proposed to train a casual CTC encoder with Gumbel-Sinkhorn network (Mena et al., 2018) to reorder the states. Zhang and Feng (2023) proposed HMT to learn when to start translating in the form of HMM, achieving the current state-of-the-art SiMT performance.

**NAR Generation** Non-autoregressive models generate tokens parallel to the sacrifice of target-side dependency (Gu et al., 2018). This property eliminates the need for teacher forcing, motivating researchers to explore flexible training objectives that alleviate strict position-wise alignment imposed by the naive MLE loss. Libovický and Helcl (2018) proposed latent alignment model with CTC loss (Graves et al., 2006), and Shao and Feng (2022) further explored non-monotonic latent alignments. Shao et al. (2020, 2021) introduced sequence-level training objectives with reinforcement learning and bag-of-ngrams difference. Ghazvininejad et al. (2020) trained NAT model using the best monotonic alignment and Du et al. (2021) further extended it to order-agnostic cross-entropy loss. In addition, some researchers are focusing on strengthening the expression power to capture the token dependency. Huang et al. (2022) proposed directed acyclic graph layer and Gui et al. (2023) introduced probabilistic context-free grammar layer. Building upon that, Shao et al. (2022) proposed Viterbi decoding and Ma et al. (2023) further explored fuzzy alignment training, achieving the current state-of-the-art NAR model performance. Apart from text translation, the NAR model also demonstrated impressive performance in diverse areas such as speech-to-text translation (Xu et al., 2023), speech-to-speech translation (Fang et al., 2023) and text-to-speech synthesis (Ren et al., 2021).

## 7  Conclusion

In this paper, we propose non-autoregressive streaming Transformer (NAST) to address the non-monotonicity problem and the source-info leakage bias in existing autoregressive SiMT models. Comprehensive experiments demonstrate its effectiveness.

## Limitations

We have observed that the performance of NAST is less satisfactory when translating from English to Romanian (En→Ro) compared to translating from German to English (De→En). This can be attributed to the fact that Romanian shares the SVO (Subject-Verb-Object) grammar with English, while German follows an SOV (Subject-Object-Verb) word order. NAST excels in handling word reordering in translating from SOV to SVO, especially there is a strict requirement for low latency. But it is relatively less effective in SVO-to-SVO translation scenarios where there is typically a monotonic alignment between the source and reference.

## Acknowledgements

We thank the anonymous reviewers for their insightful comments.

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

## A Derivation of Equation 19

We present the detailed derivation of Equation 19 in this section.

$$\mathbb{E}_{\boldsymbol{a}}[\tau_{g^{\boldsymbol{a}}}(|\boldsymbol{x}|)] = \sum_{\boldsymbol{a}} p(\boldsymbol{a}|\boldsymbol{x}) \sum_{i=1}^{(|\boldsymbol{x}|-1)\lambda} \mathbb{1}(\boldsymbol{a}_i)$$

$$= \sum_{i=1}^{(|\boldsymbol{x}|-1)\lambda} \sum_{\boldsymbol{a}} p(\boldsymbol{a}|\boldsymbol{x})\mathbb{1}(\boldsymbol{a}_i) \qquad (28)$$

$$= \sum_{i=1}^{(|\boldsymbol{x}|-1)\lambda} p(\mathbb{1}(\boldsymbol{a}_i)),$$

$$\mathbb{E}_{\boldsymbol{a}}\left[\sum_{i=1}^{(|\boldsymbol{x}|-1)\lambda} m(i)\mathbb{1}(\boldsymbol{a}_i)\right] = \sum_{\boldsymbol{a}} p(\boldsymbol{a}|\boldsymbol{x}) \sum_{i=1}^{(|\boldsymbol{x}|-1)\lambda} m(i)\mathbb{1}(\boldsymbol{a}_i)$$

$$= \sum_{i=1}^{(|\boldsymbol{x}|-1)\lambda} m(i) \sum_{\boldsymbol{a}} p(\boldsymbol{a}|\boldsymbol{x})\mathbb{1}(\boldsymbol{a}_i)$$

$$= \sum_{i=1}^{(|\boldsymbol{x}|-1)\lambda} m(i)p(\mathbb{1}(\boldsymbol{a}_i)),$$

$$(29)$$

where $p(\mathbb{1}(\boldsymbol{a}_i))$ denotes the probability that the $i$-th token in the alignment is reserved after collapsing.

## B Numerical Results with More Metrics

In addition to Average Lagging (AL; Ma et al., 2019), we also incorporate Consecutive Wait (CW; Gu et al., 2017), Average Proportion (AP; Cho and Esipova, 2016), and Differentiable Average Lagging (DAL; Arivazhagan et al., 2019) as metrics to evaluate the latency of NAST.

We adjust $l_{\min}$ in $\mathcal{L}_{\text{latency}}$ and $k$ in chunk wait-$k$ strategy to achieve varying quality-latency trade-offs. For clarity, we present the numerical results of NAST using specific hyperparameter settings in Table 3 and Table 4. Note that $\mathcal{L}_{\text{latency}}$ is applied to achieve lower latency, while the chunk wait-$k$ strategy is employed to improve translation quality. Therefore, we apply $\mathcal{L}_{\text{latency}}$ only when $k = 0$.

Table 3: Numerical results of NAST on De→En. "-" indicates that $\mathcal{L}_{\text{latency}}$ is not applied.

| | | | | | | |
|---|---|---|---|---|---|---|
| *WMT15 De→En* | | | | | | |
| $l_{\min}$ | $k$ | **CW** | **AP** | **AL** | **DAL** | **BLEU** |
| 0 | 0 | 1.44 | 0.52 | 0.65 | 1.96 | 27.73 |
| 1 | 0 | 1.51 | 0.57 | 1.87 | 3.24 | 29.82 |
| 3 | 0 | 1.60 | 0.62 | 2.97 | 4.60 | 30.46 |
| - | 0 | 1.74 | 0.66 | 4.02 | 5.89 | 30.69 |
| - | 3 | 2.03 | 0.72 | 5.83 | 7.64 | 31.58 |
| - | 5 | 2.18 | 0.75 | 6.85 | 8.39 | 31.70 |
| - | 7 | 2.59 | 0.79 | 8.44 | 9.88 | 31.94 |

Table 4: Numerical results of NAST on En→Ro. "-" indicates that $\mathcal{L}_{\text{latency}}$ is not applied.

| | | | | | | |
|---|---|---|---|---|---|---|
| *WMT16 En→Ro* | | | | | | |
| $l_{\min}$ | $k$ | **CW** | **AP** | **AL** | **DAL** | **BLEU** |
| 0 | 0 | 1.41 | 0.50 | 0.36 | 1.77 | 24.79 |
| - | 0 | 1.46 | 0.55 | 1.58 | 3.24 | 26.30 |
| - | 3 | 1.54 | 0.65 | 3.90 | 5.34 | 30.01 |
| - | 5 | 1.81 | 0.72 | 5.89 | 7.25 | 30.98 |
| - | 7 | 2.24 | 0.77 | 7.85 | 9.14 | 31.30 |

## C Case Study

To gain further insights into NAST's behavior, we examine the generation processes of two different cases within the De→En test set. We visualize the generation by plotting the generated partial alignments and the collapsed outputs at each step.

| Source | die Premierminister Indiens und Japans trafen sich in Tokio .  | | |
|---|---|---|---|
| | *the Prime Minister India and Japan met in Tokyo* | | |
| **Reference** | India and Japan prime ministers meet in Tokyo .  | | |
| **Step** | **Inputs** | **Alignments** | **Outputs** |
| 1 | die | the
the
the | the |
| 2 | Premierminister | the
the
the | |
| 3 | Indiens | the
prime
ministers | prime
ministers |
| 4 | und | of
of
India | of
India |
| 5 | Japans | India
and
and | and |
| 6 | tra@@ | and
and
Japan | Japan |
| 7 | fen | Japan
Japan
Japan | |
| 8 | sich | Japan
Japan
Japan | |
| 9 | in | Japan
Japan
Japan | |
| 10 | Tok@@ | \<blank>
met
met | met |
| 11 | io | met
\<blank>
in | in |
| 12 | . | in
in
Tokyo | Tokyo |
| 13 |  | Tokyo
.
\ | .
\ |

Figure 8: Case study of #0 in De→En test set, where we configure NAST with $l_{\min} = 1$ and $k = 0$.

In Figure 8, we illustrate a case in which NAST reorders words at the phrase-level compared to the reference. With the streaming input "*die Premierminister Indiens und Japans*", NAST produces "*the prime ministers of India and Japan*" instead of

the reference "*India and Japan prime ministers*". This output represents a source-monotonic-aligned phrase, thereby effectively reducing latency.

| Source | es sieht so aus , als könne niemand davon ausgehen , sicher zu sein .  | | |
|---|---|---|---|
| | *It seems like    as if no one could assume    safe    to be* | | |
| **Reference** | no one , it seems , can be sure that they are safe .  | | |
| **Step** | **Inputs** | **Alignments** | **Outputs** |
| 1 | es | \<blank\> 
 \<blank\> 
 \<blank\> | |
| 2 | sieht | \<blank\> 
 \<blank\> 
 \<blank\> | |
| 3 | so | \<blank\> 
 it 
 it | it |
| 4 | aus | it 
 it 
 it | |
| 5 | , | it 
 it 
 it | |
| 6 | als | it 
 it 
 it | |
| 7 | könne | looks 
 as 
 as | looks 
 as |
| 8 | niemand | as 
 as 
 as | |
| 9 | davon | as 
 if 
 if | if |
| 10 | ausgehen | no 
 no 
 one | no 
 one |
| 11 | , | one 
 one 
 one | |
| 12 | sicher | can 
 can 
 can | can |
| 13 | zu | expect 
 expect 
 to | expect 
 to |
| 14 | sein | to 
 to 
 to | |
| 15 | . | be 
 be 
 sure | be 
 sure |
| 16 |  | safe 
 . 
  | safe 
 . 
  |

Figure 9: Case study of #1083 in De→En test set, where we configure NAST with $k = 0$ and $\mathcal{L}_{\text{latency}}$ not applied.

In Figure 9, we depict another generation case where NAST manages word reorderings at the sentence level in comparison to the reference. In order to ensure low latency, NAST adjusts the sentence structure while maintaining meaning consistency with the reference. When NAST processes the source words "*es sieht so au*", it promptly generates "*it looks as if*" and continues generating the subsequent words within this grammatical structure. This ensures listeners keep synchronized with the speaker.