# OpenReview forum: "Non-autoregressive Streaming Transformer for Simultaneous Translation"
_EMNLP/2023/Conference — EMNLP 2023 Main_

### Official Review · Reviewer_JCCF · 2023-08-06

**Soundness:** 4

**Excitement:**

4: Strong: This paper deepens the understanding of some phenomenon or lowers the barriers to an existing research direction.

**Paper Topic And Main Contributions:**

his paper is about a new approach to Non-autoregressive Streaming Transformer (NAST) for Simultaneous Translation (SiMT). The paper addresses the issue of aggressive anticipation in previous autoregressive SiMT models, which results in poor translation quality and high latency. The main contribution of the paper is the development of a new NAST approach that enables flexible READ/WRITE strategy and alignment-based latency loss, which outperforms previous autoregressive SiMT models in terms of translation quality and latency. The paper presents experimental results on SiMT benchmarks, including WMT15 German → English and WMT16 English → Romanian, demonstrating the effectiveness of the proposed NAST approach.

**Questions For The Authors:**

1. The performance of De -> En and En -> Ro looks pretty different. What's the reason behind it?
2. I'm also curious about the En -> De performance since De is an SOV language and En is a SVO language. What will this language difference affect the final performance.

**Reasons To Accept:**

The paper on Non-autoregressive Streaming Transformer for Simultaneous Translation introduces a novel approach to SiMT that combines several new ideas to address the issue of aggressive anticipation in previous autoregressive SiMT models. The proposed approach includes non-autoregressive decoding, which enables intra-chunk parallelism and flexible READ/WRITE strategy. The use of repeated token or blank token to represent delay allows the model to adjust its translation speed and latency dynamically. The alignment-based latency loss encourages the model to generate translations that are non-monotonic but still accurate, which is crucial for SiMT. By combining these ideas, the proposed approach achieves state-of-the-art performance on various SiMT benchmarks, including WMT15 German → English and WMT16 English → Romanian.

**Reasons To Reject:**

The paper lacks some analysis of the results and examples to illustrate the methods

**Reproducibility:**

3: Could reproduce the results with some difficulty. The settings of parameters are underspecified or subjectively determined; the training/evaluation data are not widely available.

**Reviewer Confidence:**

4: Quite sure. I tried to check the important points carefully. It's unlikely, though conceivable, that I missed something that should affect my ratings.

**Typos Grammar Style And Presentation Improvements:**

I suggest the authors to present some generation examples to better illustrate the methods and results.

---

> ### Author Rebuttal · Authors · 2023-08-28
>
> Thank you for your constructive comments. We provide discussions and explanations about your concerns as follows.
>
> > The paper lacks some analysis of the results and examples to illustrate the methods.
>
> We appreciate your suggestions regarding the further anaysis. To provide a more comprehensive analysis of our model, we partition the De -> En test set into subsets based on the number of crosses in the alignments between the source and reference, categorizing them as easy, medium, and hard. Intuitively, a higher number of cross alignments pose greater challenges for a SiMT model. We compare our NAST with previous state-of-the-art AR SiMT model, HMT. The results are outlined below:
>
> |NAST| | | | | | | |
> | --- | --- | --- | --- | --- | ---| ---| ---|
> |AL | 0.65 |	1.87 |	2.97 |	4.02 |	5.83 |	6.85 |	8.44 |
> |Easy |	27.19 |	32.11 |	34.43 |	34.48 |	35.21 |	35.51 |	35.51|
> |Medium |	27.54 |	30.23 |	30.91 |	30.90 |	31.72 |	31.81 |	32.09|
> |Hard |	28.12 |	28.09 |	28.65 |	29.14 |	30.10 |	30.18 |	30.49|
>
> |HMT | | | | | | | |
> | --- | --- | --- | --- | --- | ---| ---| ---|
> |AL |	0.27 |	2.20 |	3.46 |	4.74 |	6.43 |	8.36 |	10.06 |
> |Easy |	23.70 |	31.89 |	33.59 |	34.72 |	34.80 |	35.44 |	35.42 |
> |Medium |	22.22 |	27.90 |	29.86 |	30.83 |	31.68 |	31.88 |	31.70 |
> |Hard |	20.42 |	25.50 |	26.99 |	28.19 |	28.96 |	29.69 |	29.75 |
>
> Compared to HMT, we observe that NAST excels particularly in handling harder cases, especially when real-time demands are higher. This highlights the superiority of NAST in addressing the non-monotonicity issue, making it more suitable for deployment in low-latency scenarios.
>
>
> > The performance of De -> En and En -> Ro looks pretty different. What's the reason behind it?
>
> Thank you for your insightful question regarding the experimental results. We would like to make a brief discussion on the following phenomena: 1. NAST significantly outperforms previous models under low-latency across datasets, while in high-latency scenarios, NAST achieves state-of-the-art results for De -> En and is slightly inferior for En -> Ro. 2. SiMT models are able to approach the performance of full-sentence translation model on De -> En. However, on En -> Ro, there remains a notable gap between all SiMT models and the offline model.
> We believe this could be attributed to several factors:
> 1. The simultaneous translation from English to Romanian involves both SVO language pairs. When latency requirements are lenient, the real-time generation difficulty is reduced. In such cases, the simultaneous translation task between two SVO languages almost degrades into a full-sentence translation task. In those conditions, the slight inferiorty of NAST could potentially reflect the gap between the full-sentence AR and NAR models.
>
> 2. The En -> Ro dataset (0.6M) is notably smaller compared to the De -> En dataset (4.5M), which can lead to the overfitting issues. Given that SiMT models rely on a prefix-to-prefix learning approach, the problem of overfitting might be more pronounced. This could possibly explain why, even under larger latencies, all SiMT models on the En -> Ro dataset still exhibit a noticeable gap from the offline model.
>
> > I'm also curious about the En -> De performance since De is an SOV language and En is a SVO language. What will this language difference affect the final performance.
>
> We would like to express our apologies. Due to time constraints and limited computational resources, we are unable to provide experimental results for the En -> De direction during the rebuttal phase. However, we intend to include these results in a subsequent version of the paper.

---

### Official Review · Reviewer_yohv · 2023-08-07

**Soundness:** 4

**Excitement:**

4: Strong: This paper deepens the understanding of some phenomenon or lowers the barriers to an existing research direction.

**Paper Topic And Main Contributions:**

The paper discusses a novel non-autoregressive model for simultaneous translation, which is one of the first attempts for this approach. In NAR architecture proposed in the paper, the mode upsample the hidden states and run CTC training on top of it to learn the monotonic alignment between source and target. During inference time, the model applied the wait-k policy. By introducing latency control, non-monotonic latency alignments, glancing and multi-stage training, the model can further improve the performance. From the numerical results, the model shows significant improvement over baselines.





**Reasons To Accept:**

1. The methods proposed in the paper are of great novelty. This is one of the first attempts on applying NAR to simultaneous translation. Not only the BLEU score, but the NAR model also shows substantial improvement over AR model on hallucination.
2. The results are strong compared with baseline model

**Reasons To Reject:**

1. More analysis on the model behavior, such as case study will help readers better understand the model.
2. It can be particularly challenging to adapt the proposed method to other translation tasks, such as multilingual translation, low resource translation, speech translation, etc., in which the AR model is significantly better the NAR model

**Reproducibility:**

4: Could mostly reproduce the results, but there may be some variation because of sample variance or minor variations in their interpretation of the protocol or method.

**Reviewer Confidence:**

5: Positive that my evaluation is correct. I read the paper very carefully and I am very familiar with related work.

---

> ### Author Rebuttal · Authors · 2023-08-28
>
> Thank you for your constructive comments. We provide discussions and explanations about your concerns as follows.
>
> > More analysis on the model behavior, such as case study will help readers better understand the model.
>
> We appreciate your suggestions regarding the case study. Following your advice, we will incorporate a case study section in the revised version of our paper. Moreover, to provide a more comprehensive analysis of our model, we partition the De -> En test set into subsets based on the number of crosses in the alignments between the source and reference, categorizing them as easy, medium, and hard. Intuitively, a higher number of cross alignments pose greater challenges for a SiMT model. We compare our NAST with previous state-of-the-art AR SiMT model, HMT. The results are outlined below:
>
> |NAST| | | | | | | |
> | --- | --- | --- | --- | --- | ---| ---| ---|
> |AL | 0.65 |	1.87 |	2.97 |	4.02 |	5.83 |	6.85 |	8.44 |
> |Easy |	27.19 |	32.11 |	34.43 |	34.48 |	35.21 |	35.51 |	35.51|
> |Medium |	27.54 |	30.23 |	30.91 |	30.90 |	31.72 |	31.81 |	32.09|
> |Hard |	28.12 |	28.09 |	28.65 |	29.14 |	30.10 |	30.18 |	30.49|
>
> |HMT | | | | | | | |
> | --- | --- | --- | --- | --- | ---| ---| ---|
> |AL |	0.27 |	2.20 |	3.46 |	4.74 |	6.43 |	8.36 |	10.06 |
> |Easy |	23.70 |	31.89 |	33.59 |	34.72 |	34.80 |	35.44 |	35.42 |
> |Medium |	22.22 |	27.90 |	29.86 |	30.83 |	31.68 |	31.88 |	31.70 |
> |Hard |	20.42 |	25.50 |	26.99 |	28.19 |	28.96 |	29.69 |	29.75 |
>
> Compared to HMT, we observe that NAST excels particularly in handling harder cases, especially when real-time demands are higher. This highlights the superiority of NAST in addressing the non-monotonicity issue, making it more suitable for deployment in low-latency scenarios.
>
> > It can be particularly challenging to adapt the proposed method to other translation tasks.
>
> As you mentioned, there is still a performance gap between NAR and AR models in these domains. However, we also notice a recent progress for NAR models in speech-related offline tasks. Given the broader application of SiMT models in scenarios involving verbal communication, we plan to further adapt our model in simultaneous speech-to-text and speech-to-speech translation in the future.

---

### Official Review · Reviewer_N9PX · 2023-08-12

**Soundness:** 3

**Excitement:**

3: Ambivalent: It has merits (e.g., it reports state-of-the-art results, the idea is nice), but there are key weaknesses (e.g., it describes incremental work), and it can significantly benefit from another round of revision. However, I won't object to accepting it if my co-reviewers champion it.

**Paper Topic And Main Contributions:**

This paper is motivated by the two issues in simultaneous machine translation (SiMT): (1) non-monotonicity problem in the source- reference mis-alignment; (2) source-info leakage bias, which encourage the model to be optimistic, leading to the issue of hallucination.

Thus this paper propose the non-autoregressive streaming Transformer (NAST), which utilize a non-autoregressive decoder  with intra-chunk parallelism, and this model is trained with a alignment based latency loss.

**Questions For The Authors:**

see "Reasons To Reject"

**Reasons To Accept:**

(1) This paper is well motivated. it starts with the issues in SiMT, and propose its framework to address the issues.

(2) Well organized paper. The method is clearly stated in detail, and the experimental results are presented with visualization, which is easy to follow.

(3) This paper proposes to use NAT decoder to alleviate the source-info leakage bias of SiMT models with autoregressive models.



**Reasons To Reject:**

(1) i am not sure whether the idea of utilizing NAT to SiMT could garantee enough contributions to be accepted. The loss function is from the literature, and the core of this paper is to use chunks to adaptively model the READ/WRITE strategy. Could the author clarify on the novelty of the proposed method?

(2) Translation quality, in the current research field, should not be measured by traditional metrics. Maybe the authors could show that NAT does not affect the fluency, coherence of the translated sentences in the SiMT scenario? One tool one could use is GPT-4 which is often used as a scoring function for machine generated contents.

**Reproducibility:**

3: Could reproduce the results with some difficulty. The settings of parameters are underspecified or subjectively determined; the training/evaluation data are not widely available.

**Reviewer Confidence:**

2: Willing to defend my evaluation, but it is fairly likely that I missed some details, didn't understand some central points, or can't be sure about the novelty of the work.

---

> ### Author Rebuttal · Authors · 2023-08-28
>
> Thank you for your constructive comments. We provide discussions and explanations about your concerns as follows.
>
> > Could the author clarify on the novelty of the proposed method?
>
> In this paper, we introduce a non-autoregressive streaming model to address the "non-monotonicity problem" and "source-info leakage" issues in previous AR SiMT models. However, designing a non-autoregressive model in the streaming scenario is not trivial. Our main contributions are concentrated on:
>
> 1. We design a non-autoregressive model architecture for streaming generation task.
>
> 2. We propose a novel alignment-based latency loss and a chunk wait-k strategy to tailor our non-autoregressive streaming model to meet various requirements of latency and quality effectively. Actually, learning when to start translating tokens to balance latency and quality is the core research problem for SiMT models.
>
> 3. We apply the non-monotonic latent alignment loss from the literature to tackle the non-monotonicity problem in SiMT.
>
> > Translation quality, in the current research field, should not be measured by traditional metrics. Maybe the authors could show that NAT does not affect the fluency, coherence of the translated sentences in the SiMT scenario? One tool one could use is GPT-4 which is often used as a scoring function for machine generated contents.
>
> We appreciate your suggestions regarding the evaluation metrics. We would like to make the discussion about the metrics in the following:
>
> 1.We firstly provide an explanation for our choice. In the literature of SiMT, most studies employ the AL-BLEU curve to showcase models' performance under various latency requirements. Hence, in order to ensure a fair and consistent comparison, our research continues with this evaluation metric.
>
> 2.To address your concerns about the fluency and coherence of non-autoregressive generation, we assess the fluency of the output on De -> En from our NAST and the previous state-of-the-art AR SiMT model, HMT. Specifically, we use the PPL value reported  by an external pre-trained language model to quantify the fluency of generation. A lower PPL value implies more fluent translations. The results are presented in the table below.
>
> |Reference| |
> |---| ---|
> |PPL | 182.37  |
>
> |NAST | | | | | | | |
> |---| ----------- | ----------- | ----------- | -----------  | ----------- | ----------- |----------- |
> |AL| 0.65	| 1.87 | 2.97 |	4.02 |	5.83 |	6.85 |	8.44 |
> |BLEU ↑| 27.73 | 	29.82	| 30.46 | 	30.69	| 	31.58 | 31.70 | 31.94	|
> |PPL ↓| 384.3 |	361.19 |	308.88	 |296.06 |	272.88 |	272.2 |	273.15|
>
> |HMT | | | | | | | |
> |---| ----------- | ----------- | ----------- | -----------  | ----------- | ----------- |----------- |
> |AL| 0.27 |	2.20	| 3.46	|4.74|	6.43|	8.36|	10.06|
> |BLEU ↑| 22.52 | 27.60 |29.29 |30.29 |30.90|31.45 | 31.61 |
> |PPL ↓| 276.43 |263.84	|245.62|	264.99 |	272.07 |	251.23 |	229.11 |
>
> We find that while NAST exhibits significantly improved translation quality compared to previous AR SiMT models, its non-autoregressive nature does impact fluency to some extent. However, we consider this trade-off acceptable. In practical scenarios like international conferences where SiMT models are employed, the language used by human speakers is often not perfectly fluent. In such contexts, the audience tends to prioritize the overall translation quality of the SiMT model under low latency, rather than the fluency of the generated sentence.
>
> 3.As far as we know, when GPT-4 is used for evaluation, it often assesses the relative quality of two outputs in a comparative manner. It is challenging to provide an absolute value of its quality, and the results are significantly influenced by the prompt used. When evaluating SiMT models, we require a metric with absolute values to assess the quality under different latencies among various SiMT models. Therefore, it becomes very hard to employ large language models to evaluate the performance of SiMT models.

---

### Meta-Review · Area_Chair_b5ej · 2023-09-17

**Recommendation:** 4

**Metareview:**

This paper proposes a non-autoregressive streaming transformer for simultaneous translation, which addresses two current problems: "non-monotonicity problem" and "source-info leakage" .  Alignment are specially considered for setting the latency loss and translation loss, which is important in the simultaneous translation setting. The results show substantial improvement over existing research. Reviewers suggest more analysis into the model behaviors.

---

### Decision · Program_Chairs · 2023-10-07

**Decision:**

Accept-Main

**Comment:**

This paper proposes a non-autoregressive streaming transformer for simultaneous translation, which addresses two current problems: "non-monotonicity problem" and "source-info leakage" .  Alignment are specially considered for setting the latency loss and translation loss, which is important in the simultaneous translation setting. The results show substantial improvement over existing research. Reviewers suggest more analysis into the model behaviors.